# Molecular Imaging Investigations of Polymer-Coated Cerium Oxide Nanoparticles as a Radioprotective Therapeutic Candidate

**DOI:** 10.3390/pharmaceutics15082144

**Published:** 2023-08-15

**Authors:** Philip Reed McDonagh, Sundaresan Gobalakrishnan, Christopher Rabender, Vimalan Vijayaragavan, Jamal Zweit

**Affiliations:** 1Department of Radiation Oncology, Virginia Commonwealth University Health System, Richmond, VA 23219, USA; 2Center for Molecular Imaging, Virginia Commonwealth University Health System, Richmond, VA 23219, USA; 3Department of Radiology, Virginia Commonwealth University Health System, Richmond, VA 23219, USA

**Keywords:** radioprotective therapeutic agent, cerium oxide nanoparticles, PET imaging, MRI, polymer coating, redox chemistry, biodistribution, pharmacokinetics

## Abstract

Cerium oxide nanoparticles (CONPs) have a unique surface redox chemistry that appears to selectively protect normal tissues from radiation induced damage. Our prior research exploring the biocompatibility of polymer-coated CONPs found further study of poly-acrylic acid (PAA)-coated CONPs was warranted due to improved systemic biodistribution and rapid renal clearance. This work further explores PAA-CONPs’ radioprotective efficacy and mechanism of action related to tumor microenvironment pH. An ex vivo TUNEL assay was used to measure PAA-CONPs’ protection of the irradiated mouse colon in comparison to the established radioprotector amifostine. [^18^F]FDG PET imaging of spontaneous colon tumors was utilized to determine the effects of PAA-CONPs on tumor radiation response. In vivo MRI and an ex vivo clonogenic assay were used to determine pH effects on PAA-CONPs’ radioprotection in irradiated tumor-bearing mice. PAA-CONPs showed excellent radioprotective efficacy in the normal colon that was equivalent to uncoated CONPs and amifostine. [^18^F]FDG PET imaging showed PAA-CONPs do not affect tumor response to radiation. Normalization of tumor pH allowed some radioprotection of tumors by PAA-CONPs, which may explain their lack of tumor radioprotection in the acidic tumor microenvironment. Overall, PAA-CONPs meet the criteria for clinical application as a radioprotective therapeutic agent and are an excellent candidate for further study.

## 1. Introduction

Despite advances in cancer treatment using radiation therapy (RT), this modality of treatment is inherently limited by damage to normal tissues surrounding the targeted cancer. Side effects from normal tissue damage can drastically decrease the quality of life of patients undergoing therapy [1,2,3,4]. Therapies that prevent collateral normal tissue damage during RT, known as radioprotectors, can significantly benefit patients by enhancing the therapeutic index and improving patient quality of life without sacrificing tumor control probability [5].

The use of radioprotectors in the clinic has been limited due to the limited number of compounds having the required properties for appropriate use in humans [6]. Firstly, radioprotectors must protect normal tissues from radiation damage while preserving radiation’s therapeutic effect on the tumor. They must also have a toxicity profile that is not worse than the expected side effects from radiation treatment. Finally, they must have an appropriate means of administration, i.e., oral or i.v., and be able to concentrate in the normal tissues that would be exposed to radiation. Very few compounds have met these criteria and have been FDA-approved for use in humans to prevent radiation induced side effects. One of these compounds is amifostine, a thiol derivative that acts as an antioxidant and may also have actions to induce hypoxia and condense DNA to reduce radiation damage, though mechanisms are not completely understood [7]. While amifostine has been shown in the clinic to decrease side effects during head and neck cancer radiation, it requires large doses and has a significant side effect profile on its own, which has prevented its widespread adoption [6,8,9].

Cerium oxide nanoparticles (CONPs) are a radioprotective therapeutic candidate with a long history of documented low toxicity and safe use in industrial and biomedical applications, including use in glass polishing, automotive catalytic convertors, UV absorption, wound dressings, and as antibacterial/antiviral agents [10,11,12,13,14,15]. CONPs have shown great promise in their application in medicine as free radical scavenging antioxidants, showing effective reduction of disease progression and protection of tissues from oxidative and inflammatory damage, as demonstrated in neuronal, ocular, heart, and numerous other tissues [16,17,18,19,20,21,22]. This antioxidant activity stems from their unique surface chemistry consisting of two oxidation states, Ce^3+^ and Ce^4+^, with oxygen vacancies, which allows them to mimic the catalytic activity of enzymes such as superoxide dismutase (SOD), catalase, oxidases, and peroxidases [22,23,24,25]. CONPs have also been shown to up-regulate the activity of antioxidant enzymes such as SOD and peroxidase [26,27]. In effect, these properties of CONPs allow them to reduce the oxidative damage that is ubiquitous in disease pathology. Conversely, CONPs have also shown intrinsic application as anti-cancer agents, producing cytotoxic effects in several tumor types related to pro-oxidative effects and affecting pathways linked to inflammation and apoptosis [22,28,29,30,31,32]. 

Among the many applications of their radical scavenging properties, one of the most promising is protecting normal tissues during radiation therapy. CONPs have been shown under many conditions to be excellent radioprotectors of several normal tissue types [26,33,34,35,36,37,38,39,40,41]. For example, in similar studies assessing radioprotection of normal cell culture lines, CONPs were able to increase the cell viability of fibroblasts by 20% following 6 MV photon irradiation to 4 Gy and of epithelial cells by 27% following 18 MV photon irradiation to 3 Gy [33,34]. Ex vivo studies have also shown this protection of normal tissue. CONPs administered intraperitoneally (i.p.) to mice prior to irradiation of the abdomen to 20 Gy using 160 kV photons had a decrease in apoptotic colon crypt cells compared to control [26]. CONPs given i.p. to mice before irradiation of the thorax to 18 Gy using 6 MV photons were able to significantly reduce neutrophile aggregation and lung tissue collapse [35]. A similar study of mice treated with i.p.-injected CONPs prior to 15 Gy thorax irradiation with 320 kV photons showed that CONPs decreased lung structural damage, collagen deposition, inflammation, and vascular damage compared to control mice [36]. In this same study, survival at 160 days from irradiation increased from 10% in untreated mice to 90% in CONP-treated mice. Another study showed that mice exposed to a lethal dose of 7 Gy photon irradiation had increased survival if they were administered CONPs i.v. or i.p. 15 min before or after irradiation, indicating CONPs may also have some radiomitigating effects [37]. These studies show the vast potential of CONPs as radioprotectors to reduce the side effects of radiation therapy.

There have also been several studies that have shown CONPs’ other important characteristic as a radioprotective therapeutic candidate, which is their lack of radioprotection of cancer cells. In the above-mentioned study, the protection of fibroblasts was compared to that of a breast cancer cell line for which CONPs showed no radioprotection [34]. CONPs not only appear to lack protection for cancer cells but may also be radiosensitizing, which has been demonstrated in pancreatic cancer and leukemia cell lines [42,43]. One possible mechanism for CONPs’ selective radioprotection of normal tissues is that CONPs’ pH-sensitive redox properties may cause them to become inactive in an acidic tumor microenvironment [44,45]. Oxidative anti-cancer effects of CONPs have also previously been shown to be pH-dependent with high activity in weakly acidic environments at pH 5.0–7.0 and negligible at a physiologic pH of 7.4 [46]. 

While CONPs show enormous potential as a radioprotective therapeutic candidate based on their radioprotective properties, a significant limitation of CONPs is their tendency to aggregate in aqueous and biofluid solutions in the absence of surface modification. In animal studies, i.v. injected uncoated CONPs show poor distribution and clearance due to this aggregation, leading to sequestration in the lungs on the first pass and significant uptake by the reticuloendothelial system with concentration in the liver and spleen, with similar results seen with i.p. and subcutaneous administration [12,47,48,49]. To create more biocompatible formulations of CONPs, many studies have shown the benefits of using a biocompatible polymer surface coating, which can prevent CONP aggregation and precipitation [44,45,46,48,50,51,52,53]. Polymer composites of CONPs have been well studied and have expanded CONPs applications to include CONP-filled scaffolds for tissue engineering, CONP infused hydrogels for wound healing/dressing, and Layer-by-Layer microcapsules containing CONPs for drug delivery [13,21]. Polymer composites can not only reduce CONP toxicity but can also allow for tunable surface properties, stimulated and targeted drug release, and improved biodistribution [21,48]. Overall, the development of CONP-polymer composites can have a synergistic effect, enhancing their potential compared to either alone [15,21,46]. 

Our prior work explored coating CONPs with several polymer coatings to reduce their size and increase their biocompatibility [48,49]. Radionuclides such as ^141^Ce and ^89^Zr for labeling to enable in vivo SPECT/PET imaging and ex vivo gamma counting to demonstrate improved biodistribution and clearance after i.v. injection in mice. Of the polymer-coated CONPs, PAA-CONPs were found to have the most promising biocompatibility in terms of biodistribution and pharmacokinetics [48]. Prior characterization of the PAA-CONPs used in the current study showed their size by TEM measurement to be ~1 nm, hydrodynamic size at ~4 nm, and zeta potential of ~−9.0 mV. By comparison, uncoated CONPs had a TEM size of ~5 nm but a hydrodynamic size of ~160 nm due to aggregation and a zeta potential of ~−22 mV. In vivo PET imaging and ex vivo biodistribution of ^89^Zr-doped CONPs revealed PAA-CONPs’ ultrasmall size allowed for renal clearance of greater than 75% at 4 h while retaining excellent distribution to normal tissues without excess lung, reticuloendothelial, or tumor uptake. 

Prior data has also addressed concerns about the interaction of PAA-coated CONPs with components of blood, which has the potential to lead to protein coating and affect CONPs’ cellular interactions [54]. Plasma samples from mice injected with ^89^Zr-doped PAA-CONPs were taken 2 h after injection and run through HPLC under conditions identical to CONPs in aqueous solution with monitoring at 254 nm. While there were some distal peaks likely representing the development of protein corona on a portion of the PAA-CONPs, the primary peak seen from PAA-CONPs in aqueous solution remained the dominant peak and was confirmed with the concurrent detection of the radioactive ^89^Zr signal [42]. These results suggest the majority of PAA-CONPs were devoid of significant protein corona after exposure in vivo, similar to a prior study of PAA-CONPs incubated in cell culture media [50]. Another study showed there was protein adsorption to PAA-CONPs in bovine serum albumin solution and cell culture media with fetal bovine serum, but this did not significantly impact their hydrodynamic size, and they showed 90% retention of SOD and catalase mimetic properties [55].

Other studies have been performed using PAA-coated CONPs that have also shown promising results. PAA-coated CONPs have been previously shown to retain enzymatic mimetic properties, reduce intracellular reactive oxygen species (ROS), and show no cytotoxic effect in osteoblasts at high doses [52]. A study of several surface-coated CONPs, including PAA, revealed no interference with their ability to react with H_2_O_2_, and smaller-diameter nanoparticles were more reactive toward H_2_O_2_ [53]. PAA-coated CONPs have also been used as a platform to functionalize CONPs with a targeting ligand using conjugation of glycine-arginine-aspartic acid (cRGD) to increase their uptake in endothelial cells, which did not impact their enzymatic properties and showed anti-inflammatory properties [56]. Based on this promising prior work, further investigation of PAA-CONPs as a radioprotective therapeutic candidate is warranted. 

As mentioned previously, prior studies have shown that polymer coated CONPs appear to retain the autocatalytic ability of uncoated CONPs. However, it remains to be demonstrated how efficacious coated CONPs are in comparison to uncoated CONPs in terms of radiation protection in vivo. Due to the immense potential of polymer-coated CONPs as a more biocompatible formulation, this work tests the radioprotective efficacy of PAA-CONPs compared to uncoated CONPs and the established radioprotective agent amifostine. In this study, we selected normal colon tissue to test these radioprotective properties due to its high and acute radiosensitivity related to damage to mucosal stem cells in the crypts of Lieberkuhn [1,26].

To further explore the mechanisms of CONPs radioprotection, and especially their lack of protection against tumors, we designed experiments to test their redox properties at varying pH. By testing PAA-CONPs’ pH-dependent redox properties in aqueous solution as well as in vivo, we aim to further characterize the mechanisms by which CONPs may preferentially protect normal tissues. We expect this better understanding of CONPs’ mechanisms of action will help inform future studies and determine the most appropriate clinical applications for CONPs.

## 2. Materials and Methods

### 2.1. Synthesis of Coated and Uncoated CONPs

The synthesis of coated and uncoated CONPs was completed using methods described previously [48]. All reactants were purchased from Sigma-Aldrich, St. Louis, MO, USA. Briefly, CONPs with and without polymer coating were synthesized in a single pot co-precipitation reaction with a solution of cerium(III) nitrate hexahydrate (Ce(NO_3_)_3_·6H_2_O, 16.7 mM, 150 µL) added to a solution of ammonium hydroxide (NH_4_OH, 1.45 M, 150 µL). Prior to addition to ammonium hydroxide, poly(acrylic acid) (PAA, Mw ~1800, 2.5 mg) was added to the cerium salt solution for polymer coating, while no additions were made for uncoated CONPs. The solutions were stirred for >20 h at room temperature, and unreacted reactants were removed by filtering four times through 30,000 and 3000 MWCO filters (MilliporeSigma, Burlington, MA, USA) for uncoated and PAA coating, respectively. The cut-off solution was collected and diluted with deionized water. Concentration was measured by inductively coupled plasma-optical emission spectrometry (ICP-OES) measurement on a Vista-MPX CCD Simultaneous ICP-OES (Varian, Palo Alto, CA, USA).

### 2.2. Spontaneous Colon Tumor Model

All animal experiments were performed according to the policies and guidelines of the Animal Care and Use Committee (IACUC) at Virginia Commonwealth University, and the procedures followed were in accordance with institutional guidelines and humane care. Colorectal cancer was induced in adult female C57BL/6 mice (NCI, Rockville, MD, USA). Mice were i.p. injected with 15 mg/kg of the carcinogen azoxymethane, followed by three cycles of colitis induction with DSS in drinking water (2% *w*/*v*) over 10 weeks, as previously described [57].

### 2.3. Mouse Irradiation

Two systems were used for the irradiation of mice. A Cesium-137 (^137^Cs) source was used for whole body mouse irradiation (Gammacell 40 research irradiator, MDS Nordion, Toronto, ON, Canada), with mice receiving a single whole body dose of 10 Gy. The small animal radiation research platform (SARRP, Xstrahl Inc., Suwanee, GA, USA) was used for localized stereotactic mouse irradiation to 10 Gy in a single fraction. SARRP irradiation of the mouse colon was performed on anesthetized mice (2% isoflurane in oxygen) using 220 kV photons at 13 mA targeted to the midline mouse colon. Onboard imaging using cone beam CT (65 kV, 1 mA) allowed for tissue density calculation for dosimetry and anatomic delineation of tissues. Beams were targeted from the mouse rectum to 3 cm cranial using 2 orthogonal beam positions at each of 3 isocenters 1 cm apart using a 1 cm square collimator. The first isocenter was selected to cover the entire rectum, and each subsequent isocenter was placed exactly 1 cm cranially. Two orthogonal beams were used at each beam location at 45° and −45° to prevent high surface dose deposition and avoid attenuation of the beam by the spine. SARRP irradiation of xenograft tumor-bearing mice with tumors on their rear flanks was performed on anesthetized mice using a single beam position at 0° and a 1 cm square collimator.

### 2.4. CONP, Amifostine, and Sodium Bicarbonate Treatment

CONPs were administered to mice at a dose of 1 mg/kg (cerium mass) in 200 μL saline by tail vein injection. Amifostine (Sigma-Aldrich, St. Louis, MO, USA) was administered to mice at a dose of 400 mg/kg in 200 μL of saline by tail vein injection. This dosing was based on a prior clinical trial that showed protection by amifostine in non-Hodgkin’s lymphoma patients who were administered a dose of 740 mg/m^2^, which approximately converts to 400 mg/kg in mice [9]. For irradiated mice, CONPs and amifostine were injected 2 h prior to treatment. Sodium bicarbonate was administered to mice with 700 µL of 1 M sodium bicarbonate solution by oral gavage. Based on the response of tumor pH in the literature, mice were administered sodium bicarbonate three hours prior to irradiation [58].

### 2.5. Ex Vivo Normal Colon Tissue Analysis

A fluorescent terminal deoxynucleotidyl transferase dUTP nick end labeling (TUNEL) assay (ApopTag Fluorescein In Situ Apoptosis Detection Kit, MilliporeSigma, Burlington, MA, USA) was used to measure apoptosis in mouse colon tissue. Mice were separated into groups of control, irradiation only (*n* = 3), PAA-CONP pre-treatment plus irradiation (*n* = 6), and uncoated CONP pre-treatment plus irradiation (*n* = 3). Amifostine pre-treated animals were also irradiated to compare CONPs to an established radioprotective drug (*n* = 3). Following SARRP irradiation of the mouse colons, all mice were sacrificed at 4 h based on prior research showing this time point has maximally evident apoptosis in mouse colons [59]. The mouse colons were then excised and frozen in an optimal cutting temperature (OCT) compound as a “swiss roll”, as previously described [60]. Colon cryosections were created at 6 μm on a cryostat (Leica Biosystems, Deer Park, IL, USA), placed on a microscope slide, and preserved at −20 °C until staining. The slides were stained using the ApopTag assay and counterstained with DAPI (MilliporeSigma, Burlington, MA, USA) for nuclear staining in Vectashield mounting media (Vector Laboratories, Inc., Newark, CA, USA). Analysis of colon slides was completed in ImageJ 1.54d (NIH, Rockville, MD, USA) by using threshold quantification of staining area after artifact removal. The ratio of apoptosis staining to nuclear staining was used to determine the apoptotic index.

### 2.6. In Vivo PET/CT [^18^F]FDG Imaging of Spontaneous Colon Tumors

To evaluate the effects of CONPs on tumor response to radiation, [^18^F]FDG PET imaging was used before and after SARRP irradiated spontaneous colon tumor bearing mice using a pre-clinical PET/CT system (Siemens Healthcare, Malvern, PA, USA). Mice were separated into groups of control, irradiation only, PAA-CONP pre-treatment plus irradiation, or uncoated CONP pre-treatment plus irradiation (*n* = 3). An initial scan of spontaneous colon tumor-bearing mice provided a baseline of metabolic activity in the tumors. One day after [^18^F]FDG PET imaging, the colons were irradiated on the SARRP. Two weeks after irradiation, [^18^F]FDG PET imaging was repeated to determine tumor response.

Analysis of [^18^F]FDG PET images was completed in Inveon Research Workplace 4.1 (Siemens Healthcare, Malvern, PA, USA). Regions of interest (ROIs) were created in the coronal plane outlining the colon tumors, careful to avoid inclusion of the bladder due to the high urine content of [^18^F]FDG. Tumor volumes were determined using a threshold standardized uptake value (SUV) of 2.5 for all images, which corrects for the injected dose and body weight of the animal and excludes surrounding [^18^F]FDG tissue uptake. 

### 2.7. CONP Autocatalytic Activity

To determine the effect of pH on the autocatalytic activity of PAA-CONPs, solutions were prepared at 0.5 mg/mL in buffered solutions of pH 7.4, 7.0, and 6.6. Using previously described methods, the solution absorption at wavelength 420 nm was measured before and immediately following the addition of 7.5 mM H_2_O_2_ using a multiwell plate reader (Beckman Coulter DTX880, Brea, CA, USA) (*n* = 4) [48]. Increased optical density (O.D.) at 420 nm wavelength is an indicator of the presence of stable ceric peroxides on the surface of CONPs [22,51]. Therefore, the change in optical density (O.D.) at 420 nm after reaction with H_2_O_2_ is a surrogate for the redox capacity of the nanoparticles, with lower O.D. indicating decreased ability to react with H_2_O_2_.

### 2.8. Xenograft pH Measurement

Athymic nude mice (NCI, Rockville, MD, USA) were implanted subcutaneously with 2.5 million HCT-116 colon cancer cells in matrigel in their rear flanks. A micro pH electrode probe with a needle tip (ORION^®^ Needle Tip Micro Combination pH Electrode, Thermo Fisher Scientific, Waltham, MA, USA) was used to determine the pH of the xenograft tumors with and without sodium bicarbonate pre-treatment (*n* = 4). Mice were anesthetized (2% isoflurane in oxygen) and the needle probe was inserted into the tumors for direct measurement of pH. Mice were sacrificed immediately following the measurement of pH.

### 2.9. MRI Imaging of Xenograft Tumors

A 7 T pre-clinical MRI scanner (Bruker, Billerica, MA, USA) was used to take T2-weighted scans using a 72 mm volume coil of HCT-116 xenograft bearing athymic mice before and 2 weeks following irradiation under several conditions. During scans, mice were anesthetized with 2% isoflurane in oxygen, and cardiac and breath rate monitoring was used to monitor vitals. Mice were separated into groups of control, irradiation only, sodium bicarbonate pre-treatment plus irradiation, PAA-CONP pre-treatment plus irradiation, and sodium bicarbonate and PAA-CONP pre-treatment plus irradiation (*n* = 3). Using ImageJ and the plugin VolumEst, the tumor volume was determined, and the percent change in volumes before and after irradiation was calculated [61].

### 2.10. Clonogenic Assay of Xenograft Tumors

An ex-vivo clonogenic assay of HCT-116 xenografts was used to determine tumor response to irradiation. Xenograft bearing mice were separated into groups of control, PAA-CONP treatment only, irradiation only, sodium bicarbonate pre-treatment plus irradiation, PAA-CONP pre-treatment plus radiation, and sodium bicarbonate and PAA-CONP pre-treatment plus irradiation (*n* = 3). Mice were given a whole-body radiation dose of 10 grays from the ^137^Cs source. Immediately after irradiation, tumors were excised, minced, and incubated in a solution of 0.25% trypsin and 0.002% DNase I in FBS-free DMEM/high glucose media at 37 °C for 20 min. After incubation, cells were pelleted, counted, and plated at 500 or 5000 cells per dish in 10 cm^2^ dishes for non-irradiated and irradiated tumors, respectively. Dishes were incubated for 14 days for colony growth, after which dish media was removed and the colonies were fixed with methanol/acetic acid for 10 min and stained with crystal violet (0.5%). Colonies of greater than 50 cells were counted, and plating efficiency was calculated for each dish. Percent cell survival was calculated based on the plating efficiency of untreated xenografts, which was normalized to 100%.

### 2.11. Statistical Methods

Statistical analysis was performed using a two-tailed Student’s *t*-test. A *p* value of <0.05 is considered statistically significant. Microsoft Excel 2016 (Microsoft, Redmond, WA, USA) was used for all statistical analyses.

## 3. Results

### 3.1. Normal Mouse Colon Response to Irradiation with PAA-CONP, Uncoated CONP, and Amifostine Pre-Treatment

Apoptotic indices were calculated for normal colon tissue from untreated control mice, irradiated mice, and mice pre-treated with PAA-CONP, uncoated CONP, and amifostine prior to irradiation and are presented in Figure 1. The apoptotic index was significantly increased in all irradiated animals compared to non-irradiated controls (*p* < 0.01). Compared to irradiation alone, all of the pre-treatments with PAA-CONP, uncoated CONP, and amifostine showed similar significant decreases in the apoptotic index of over 50% (*p* < 0.05).

### 3.2. [^18^F]FDG PET Imaging of Colon Tumor Response to Irradiation with PAA-CONP and Uncoated CONP Pre-Treatment

Changes in the volume of spontaneous colon tumors over the course of 2 weeks were calculated based on [^18^F]FDG PET imaging of untreated control mice, irradiated mice, and mice pre-treated with PAA-CONP and uncoated CONP prior to irradiation. Figure 2 shows that irradiation caused significant tumor regression in all animals compared to control (*p* < 0.05). Tumor volume changes were not statistically significant between irradiated mice versus either of the CONP treatments, though uncoated CONPs showed a trend toward poorer tumor control (*p* = 0.07 vs. irradiation alone).

### 3.3. pH-Dependent Redox Reaction of PAA-CONP

PAA-CONPs in buffered solutions of pH 7.4, 7.0, and 6.6 demonstrate decreased redox capacity with decreasing pH and are presented in Figure 3. PAA-CONPs showed a significant 7.6% decrease in O.D. (surrogate for redox capacity) at pH 7.0 versus pH 7.4 (*p* < 0.05) and a further significant decrease of 48% at pH 6.6 versus pH 7.4 (*p* < 0.00005).

The efficacy of sodium bicarbonate oral gavage to change tumor pH was tested by using a needle pH electrode to probe HCT-116 xenograft tumors on untreated control mice and mice pre-treated with sodium bicarbonate oral gavage. Tumor pH values for untreated mice ranged from 6.57 to 7.19, with an average pH of 6.93. Tumor pH for mice pre-treated with sodium bicarbonate ranged from 7.20 to 7.33, with an average pH of 7.24, significantly higher than the pH of tumors from untreated mice (*p* < 0.00001).

### 3.4. Sodium Bicarbonate and PAA-CONP Pre-Treatment Effects on Irradiated Xenograft Tumors

A comparison of the change in tumor volume of HCT-116 xenograft tumors, measured by MRI, between untreated control, and several irradiation conditions is presented in Figure 4. Irradiation caused decreased growth in all groups, though this was statistically significant only for tumors treated with irradiation alone and those pre-treated with sodium bicarbonate alone or PAA-CONP alone followed by irradiation (*p* < 0.05). When combined, sodium bicarbonate treatment followed by PAA-CONP treatment and then irradiation showed a trend toward increased tumor growth compared to the other irradiation groups, though this was not statistically significant.

A comparison of cell survival of HCT-116 xenograft tumors between untreated control and several irradiation conditions is presented in Figure 5. PAA-CONP-treated xenograft tumors showed no significant effect of PAA-CONP on tumor growth versus control (not shown, *p* = 0.21). Irradiated xenografts showed a significant decrease in cell survival under all conditions compared to control (*p* < 0.0005). For irradiated xenografts, both sodium bicarbonate and PAA-CONP pre-treatment caused a further decrease in cell survival compared to irradiation alone (*p* < 0.05). When mice were pre-treated with both sodium bicarbonate and PAA-CONP, this decrease was absent, the tumor showed similar survival as irradiation alone (*p* = 0.41), and the survival was significantly increased compared to both sodium bicarbonate alone and PAA-CONP alone (*p* < 0.05).

## 4. Discussion

Expanding on prior work that showed the increased biocompatibility of PAA-CONPs, this work explores the radioprotective properties of PAA-CONPs compared to uncoated CONPs and the established radioprotective therapeutic agent amifostine. Both uncoated CONPs and PAA-CONPs were able to significantly reduce apoptosis in irradiated normal mouse colons with efficacy comparable to amifostine. These results indicate the PAA-coating does not interfere with the surface redox properties of CONPs, and PAA-CONPs are able to yield the same efficacy for radioprotection as uncoated CONPs. Being the most critical aspect of a radioprotective drug, the protection of normal tissue, it is encouraging to see that both formulations showed this property at the same level of radioprotection as amifostine. Notably, this effect was seen at a much lower total dose of CONPs (1 mg/kg of cerium weight) compared to amifostine (400 mg/kg).

The level of radioprotection observed with uncoated CONPs was comparable to PAA-CONPs. This is slightly surprising based on prior biodistribution results of uncoated CONPs and PAA-CONPs, with PAA-CONPs showing uptake in mouse colons that is over 5 times greater than uncoated CONPs at the same administered dose [48]. However, this may be due to a threshold dose for protection that is significantly lower than that administered in this study. Further testing of radioprotection across several doses would be needed to determine a dose-response curve and ascertain if PAA-CONPs can outperform uncoated CONPs at lower doses.

[^18^F]FDG PET imaging of spontaneous colon tumors before and after irradiation with PAA-CONPs and uncoated CONPs pre-treatment showed that both did not prevent tumor regression after irradiation. However, pre-treatment with uncoated CONPs resulted in less tumor regression than pre-treatment with PAA-CONPs or irradiation alone. While not statistically significant, this trend may indicate some radioprotection by uncoated CONPs, which may impact tumor control and clinical outcomes. PAA-CONP treatment, however, showed the same level of tumor regression compared to irradiation alone, indicating it showed no protection of the tumor. This property makes them ideal for clinical application to ensure their action as a radioprotective drug does not adversely affect tumor control during radiation therapy.

A study of the pH-related redox properties of PAA-CONPs revealed that a modest change in pH environment causes PAA-CONPs to have a significant decrease in their ability to react with radicals in solution. With this drastic decrease in redox capacity in solution, it is reasonable to expect that CONPs have their redox properties compromised in an acidic tumor microenvironment. Indeed, the pH range seen on probing of HCT-116 xenografts demonstrates the tumor microenvironment can reach pH as low as 6.6, a pH that appears to reduce CONPs’ redox capacity by nearly 50% in solution.

pH normalization with sodium bicarbonate treatment was validated on pH probe tumor measurements, which showed tumors reached a near physiologic pH of over 7.2 compared to a pH of 6.9 in untreated tumors. MRI tumor volume measurements before and 2 weeks after irradiation revealed a trend that a pH-normalized tumor microenvironment seems to prevent tumor control at the same level as irradiation without manipulation of the pH of the tumor microenvironment. This may indicate that the normalized pH allows PAA-CONPs to regain some of their radioprotective properties, leading to poorer tumor control after irradiation. The lack of statistical significance when comparing MRI tumor volume changes with and without bicarbonate pre-treatment is likely due to the confounding of the volume measurements by the presence of edema and necrosis. Edema was seen in the 2 week follow up images of all irradiated tumors and can be seen as bright white in the T2-weighted image in Figure 4A at 2 weeks after irradiation. The likely presence of necrosis in the tumors after irradiation would also contribute to the total volume calculation. Both edema and necrosis would add to the calculated tumor volume, but not represent viable tumors, making it more difficult to extract the true tumor response from the data. 

By also performing this pH normalization study using ex vivo clonogenic plating, the limitations of MRI were eliminated, and statistically significant trends were able to be identified. The cell survival data confirms that the trend seen on MRI of PAA-CONPs regaining radioprotection at a normalized pH is statistically significant. Pre-treatment with both bicarbonate and PAA-CONPs led to increased cell survival compared to either treatment alone. These results further demonstrate the likelihood of pH playing a significant role in the mechanisms behind CONPs lack of radioprotection of tumor tissue. While there are likely more mechanisms involved in the lack of radioprotection of tumor tissue by CONPs, these results show that further study of the effects of pH change on CONPs’ radioprotection is warranted to further elucidate their mechanism of action. 

Interestingly, the clonogenic plating assay also showed that both sodium bicarbonate and PAA-CONP produced radiosensitization, with both being associated with significantly decreased cell survival compared to irradiation alone. Radiosensitization by bicarbonate exposure may warrant further study, as bicarbonate treatment has been previously shown to be associated with decreased metastases in a prostate cancer mouse model, though not in melanoma [62]. As mentioned earlier, radiosensitization by CONPs has been demonstrated previously in pancreatic cancer and leukemia cell lines [42,43]. The absence of radiosensitization CONPs in the [^18^F]FDG PET data from irradiated spontaneous colon tumors may indicate this property is only present in certain tumor types or under certain tumor microenvironment conditions. A study of CONPs anchored to graphdiyne and used in cell culture and xenograft tumors has suggested that O_2_ production from CONPs’ catalase-mimic activity may alleviate hypoxia and increase radiation effectiveness in tumors [63]. Further study will still be needed to further determine the CONP properties and tumor characteristics that can lead to reliable radiosensitization by CONPs. 

## 5. Conclusions

This research suggests that PAA-CONPs have similar radioprotective efficacy as the established radioprotective therapeutic agent amifostine by showing equivalent protection of normal mouse colon tissue. This work also demonstrates that PAA-CONPs do not show protection of tumor tissue in both spontaneous and xenograft colon tumor models using in vivo [^18^F]FDG PET and MR imaging and an ex vivo clonogenic assay. This lack of radioprotection in tumors seems to be related to their reduced redox potential when exposed to the lower pH of the tumor microenvironment. Building on prior research, the results of this study demonstrate that PAA-CONPs appear to meet all the criteria of an ideal radioprotective therapeutic agent. While further toxicity studies are warranted, prior research has shown CONPs have a limited toxicity profile, and the dose of CONPs used in this study was well tolerated with no clinically evident adverse effects seen in treated mice. Further dose-response investigation is also warranted, but our study shows that CONPs exhibited their radioprotective effects at a dose much less than the dose of amifostine used, which was chosen to be comparable to the therapeutic dose in humans. With their low-dose efficacy, excellent biodistribution, and favorable pharmacokinetics with rapid renal clearance, the overall toxicity of PAA-CONPs may be much less than that seen with amifostine while affording the same radioprotective effects and not interfering with tumor control during radiation therapy. Overall, PAA-CONPs appear to be excellent radioprotective therapeutic candidates worthy of future study. 

## Figures and Tables

**Figure 1 pharmaceutics-15-02144-f001:**
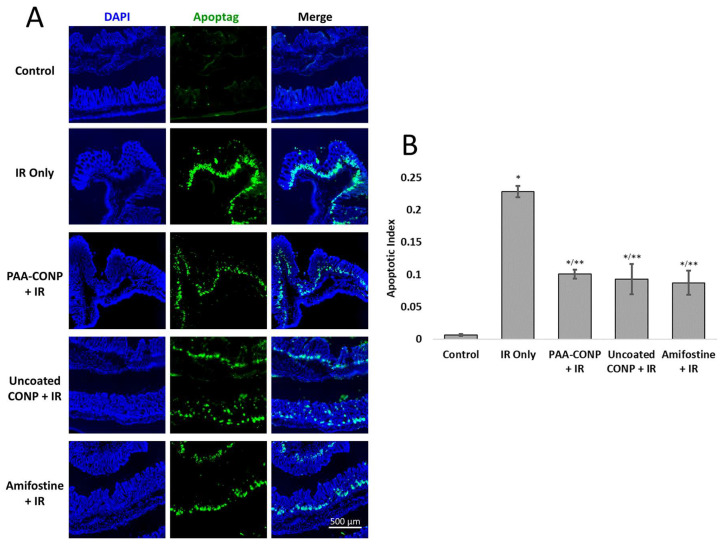
Effect of CONPs and amifostine on apoptosis in irradiated mouse colons. (**A**) Fluorescent imaging of mouse colons stained for nucleus (DAPI) and apoptosis (ApopTag) reveals the localization and extent of apoptosis. Control colons show little apoptosis staining, while irradiated colons show significant apoptosis in the region of colon crypt stem cells. (**B**) Calculation of the apoptotic index reveals low-level apoptosis in control colons, the highest apoptosis in the irradiated colons, and significant decreases in apoptosis in the colons of mice pre-treated with CONPs or amifostine, indicating significant radioprotection under these conditions. (Mean ± SEM, *n* = 3–6, * *p* < 0.05 vs. Control, ** *p* < 0.05 vs. IR Only).

**Figure 2 pharmaceutics-15-02144-f002:**
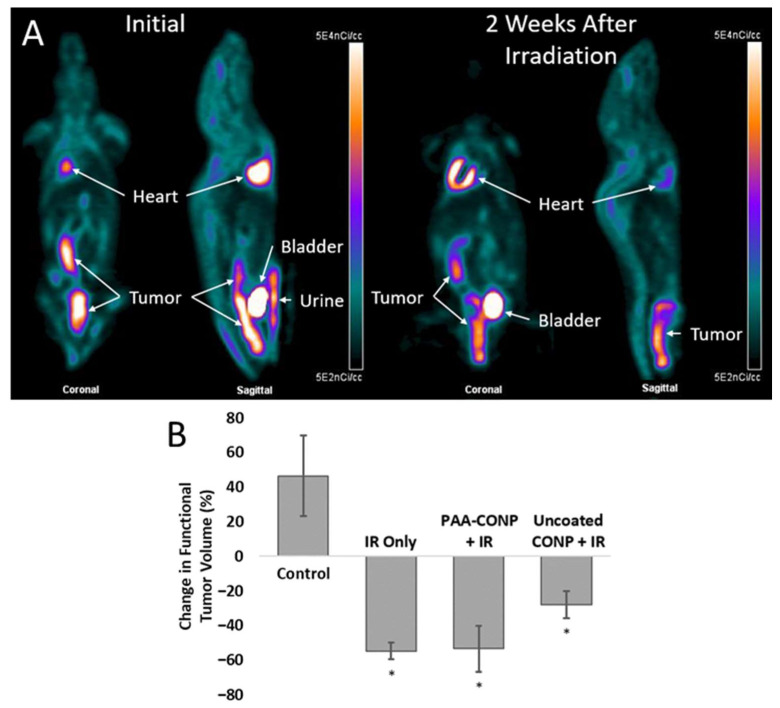
[^18^F]FDG PET measured effects of CONPs on spontaneous colon tumor response to radiation. (**A**) Representative [^18^F]FDG PET images of a mouse bearing spontaneous colon tumors before and 2 weeks after stereotactic irradiation of the colon to 10 Gy. Images demonstrate the decreased [^18^F]FDG uptake in tumors after irradiation, indicating decreased metabolic activity and response to irradiation. (**B**) Irradiated tumors showed a decrease in functional tumor volume, but only PAA-CONP pre-treatment showed the same level of tumor control compared to irradiation alone. (Mean ± SEM, *n* = 3, * *p* < 0.05 vs. Control).

**Figure 3 pharmaceutics-15-02144-f003:**
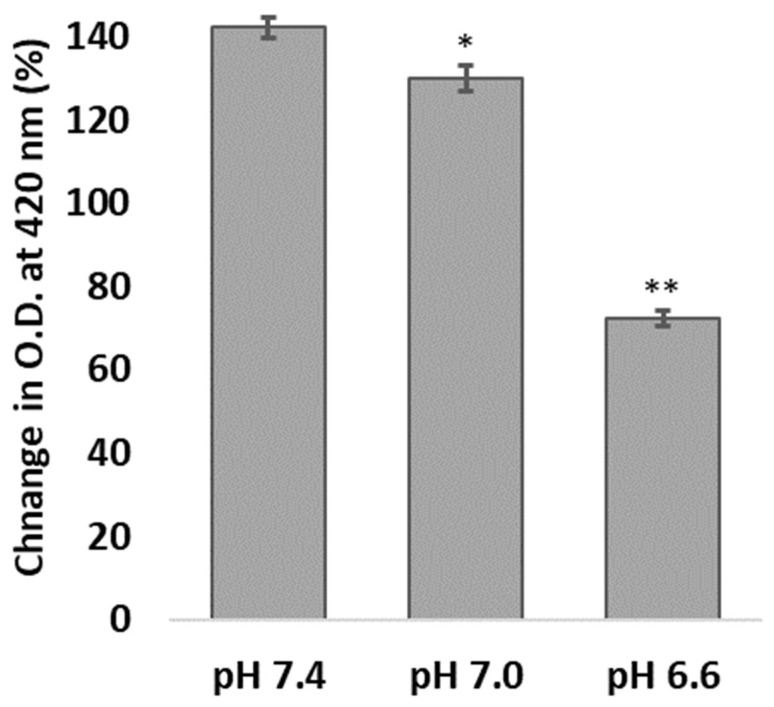
Impact of pH on PAA-CONPs’ redox reaction with H_2_O_2_. PAA-CONPs’ absorbance, monitored at 420 nm, was measured before and after addition of H_2_O_2_ in pH buffered solutions of pH 7.4, 7.0, and 6.6 to determine redox capacity (increased O.D. at 420 nm indicates formation of stable surface ceric peroxides). Compared to pH 7.4, PAA-CONPs showed decreased redox capacity at pH 7.0 and an even further decrease at pH 6.6, indicating a strong correlation between pH and CONP redox capacity. (Mean ± SEM, *n* = 4, * *p* < 0.05 vs. pH 7.4, ** *p* < 0.00005 vs. pH 7.4).

**Figure 4 pharmaceutics-15-02144-f004:**
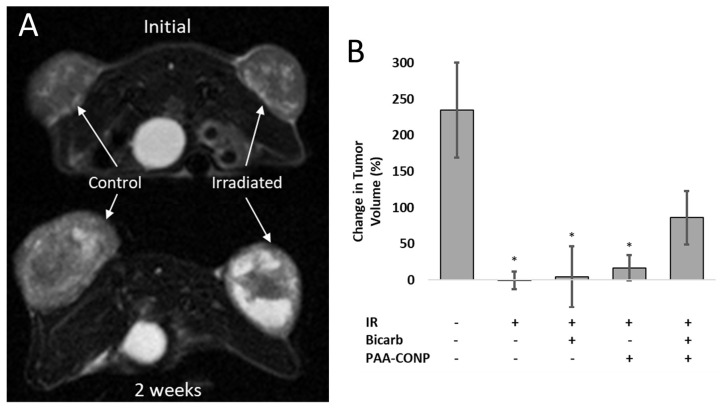
Effects of PAA-CONP and bicarbonate administration on tumor volume as measured from T2-weighted MRI. (**A**) Representative T2-weighted MRI slices from a mouse with HCT-116 colon cancer xenografts on the right and left flanks with the left flank irradiated show less tumor growth and increased edema (white) in the irradiated tumor at 2 weeks. (**B**) Tumor volumes were measured in ImageJ from MRI scans prior to and 2 weeks following irradiation using the plugin VolumEst. While tumors irradiated under all conditions showed a decrease in tumor growth compared to control, this was not statistically significant when mice were co-administered bicarbonate and PAA-CONP. (Mean ± SEM, *n* = 3, * *p* < 0.05 vs. Control).

**Figure 5 pharmaceutics-15-02144-f005:**
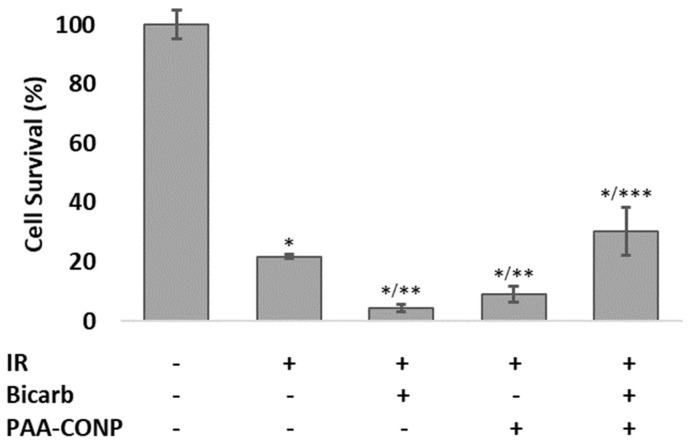
Effects of PAA-CONP and bicarbonate administration on the cell survival of irradiated xenograft HCT-116 tumors. Irradiated tumors showed significantly decreased cell survival compared to controls. Tumors from mice treated with PAA-CONP or bicarbonate before irradiation showed a further decrease in cell survival compared to tumors that were only irradiated. When treated with both bicarbonate and PAA-CONP, tumors showed an increase in cell survival compared to irradiated tumors treated with either alone. (Mean ± SEM, *n* = 3, * *p* < 0.0005 vs. Control, ** *p* < 0.05 vs. IR Only, *** *p* < 0.05 vs. IR + Bicarb and IR + PAA-CONP).

## Data Availability

No publicly archived datasets were produced from the data collected for this study. Original data are available from the corresponding author upon request.

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
