# Peer review of "Molecular Imaging Investigations of Polymer-Coated Cerium Oxide Nanoparticles as a Radioprotective Therapeutic Candidate"

_pharmaceutics, 2023, doi:10.3390/pharmaceutics15082144_

Round 1

Reviewer 1 Report

Radioprotectors are an important component of tumor radiation therapy, wherein nanosized cerium oxide seems to be a very interesting object in oncology. The presented work is timely and quite interesting, it can be published after a minor revision.

Line

Text

Remark

3

 as a Radioprotective Therapeutic

Missing noun. Probably (according to the Conclusions), “as a Radioprotective Therapeutic Candidate”?

99

cerium(III) nitrate hexahydrate (Ce(NO3) 3·6H2O) dissolved in a solution of ammonium hydroxide (NH4OH).

It is impossible to dissolve cerium nitrate in ammonia solution, since the cerium salt is immediately hydrolyzed in alkali media. Obviously, a solution of cerium salt (what is concentration, quantity?) was added to an ammonia solution (what is concentration?) It is necessary to describe the synthesis process more clearly.

180

Increased optical density (O.D.) at 420 nm wavelength is an indicator of the presence of Ce4+ on the surface of CONPs, which is converted from Ce3+ through a redox reaction with H2O2.

The absorption of cerium compounds following addition of hydrogen peroxide at 420 nm wavelength corresponds not to tetravalent cerium, but to cerium peroxide compounds.

As an example, see https://doi.org/10.1016/B978-0-12-815661-2.00008-6, Section 8.2: the scheme “colorless ceria sol + H2O2 → brown ceria sol → colorless ceria sol” does not mean the process of oxidation-regeneration of cerous ions in the nonstoichiometric ceria nanoparticles, Ce3+ + H2O2 → Ce4+ → Ce3+, but associates with the formation and decomposition of dark-colored ceric peroxides: Ce4+ + H2O2 → Ce4+(OOH) → Ce4+ (for details see Section 8.4.1.2 and Fig. 8.5).

268

increased O.D. at 420 nm indicates a change in oxidation state of surface cerium from Ce3+ to Ce4+

The same remark. Revisit the concept.

For another example, see https://doi.org/10.1021/acsami.0c08778, Fig. 4b and following text: H2O2 molecules reacts with cerium oxide nanoparticles to form a stable peroxo and/or hydroperoxo species at the particle surface. We propose here that the coordination sites for peroxide species are the critical factors in the ceria anti-oxidation process with H2O2.

257

[A] Representative [18F]FDG PET image of mouse bearing spontaneous colon tumor.

This image is duplicated from Philip R. McDonagh’s dissertation "Radioprotective Cerium Oxide Nanoparticles: Molecular Imaging Investigations of CONPs' Pharmacokinetics, Efficacy, and Mechanisms of Action", 2016 (Figure 30), see attached. It is necessary to refer to the source material and indicate this figure as an analogue. But the best way is to provide an up-to-date fresh image from this study.

Author Response

Dear Reviewer,

Thank you for your constructive comments on our manuscript.

Regarding Comment 1, we have adjusted the title, as suggested, to include "Candidate." (3-4)

Regarding Comment 2, more details have been added to the synthesis of CONPs in the Methods section, including the addition of concentrations and amounts of reactants. (176-178, 180)

Regarding Comments 3 and 4, we have corrected the text to reflect the formation and detection of ceric peroxides per the references provided. (261-263, 362-363)

Regarding Comment 5, the PET images have been updated. Rather than highlighting the method of measurement of FDG activity, for which an image from the referenced thesis was previously used, the figure now demonstrates the change in FDG activity seen after irradiation of the colon tumors. (351, 353-356)

We hope that these changes have adequately addressed your concerns. We believe these changes have significantly strengthened the paper and we are very appreciative of the feedback.

Sincerely,

Reed McDonagh

Reviewer 2 Report

The paper reports on the comprehensive study of selective radioprotection properties of cerium oxide nanoparticles (CONPs) in comparison with the well studied radioprotector, amifostine. The study was performed in vivo on a model of irradiated mouse colon to determine the effects of CONPs on normal colon and on tumor. CONPs were shown to be an excellent candidate therapeutic for the radiation therapy of cancer.
The following issues should be addressed before acceptance of this paper:

1. In the Introduction section, please provide references to the state-of-the-art papers and reviews dealing with the synthesis and properties of CeO2-polymer composites. 
Similarly, the Introduction section lacks a comprehensive review of existing papers on radioprotective properties of CeO2 nanoparticles.
Preparation of PAA-coated CeO2 nanoparticles was already reported by many research groups. Please discuss the corresponding background.

2. The data on the radiation protection provided by CONPs should be discussed along with the existing data on X-ray protective properties of cerium oxide nanoparticles.
3. Some typos are to be corrected (Line 229, idex instead of index), etc.
4. When entering the blood flow, non-coated CONPs (and, possibly, PAA-coated CONPs) would interact with the components of blood. For instance, protein corona is reportedly formed on CeO2 particles strongly affecting their stability and behaviour. Please discuss the data obtained taking this issue in mind.
5. No information is provided on the particle size, zeta potential and the other physical properties of CONPs.

Author Response

Dear Reviewer,

Thank you for your constructive comments on our manuscript.

Regarding Comment 1 and 2, we have updated our introduction to better detail the prior research regarding CONP-polymer composites and their biomedical applications, as well as expanded the sections regarding the antioxidant, anti-cancer, and radioprotective properties of CONPs. We have also added further references to prior research specific to PAA-coated CONPs. (52-155)

Regarding Comment 3, we have corrected the identified typo. (310)

Regarding Comment 4, we have previously studied the effects of exposure of PAA-CONPs to the bloodstream in mice by studying their retention time using HPLC. This data was included in the supplement in our prior paper and has been detailed in the text of the introduction to address the concerns about the formation of protein corona. Similar to prior studies that determined PAA-CONPs do not significantly change hydrodynamic size after exposure to cell culture media, PAA-CONP retention time on HPLC did not significantly change after exposure to the bloodstream in mice. While one study did show significant protein adsorption on PAA-CONPs, this did not significantly interfere with their hydrodynamic size or redox capabilities. (129-142)

Regarding Comment 5, the characterization studies of PAA-CONPs were performed in our previous study and are summarized in the introduction. Because we have used the same synthesis for this study as in our prior study, the characterization results for PAA-CONPs and uncoated CONPs used in this study are expected to be identical to those of our prior work. (121-128)

We hope that these changes have adequately addressed your concerns. We believe these changes have significantly strengthened the paper and we are very appreciative of the feedback.

Sincerely,

Reed McDonagh